# Blaschke Product Neural Network (BPNN): A Physics-Infused Neural Network for Phase Retrieval of Meromorphic Functions

**Juncheng Dong, Simiao Ren, Yang Deng,**
**Omar Khatib, Jordan Malof, Mohammadreza Soltani,**
**Willie Padilla & Vahid Tarokh**
Department of Electrical and Computer Engineering
Duke University
Durham, NC 27708, USA

## Abstract

Numerous physical systems are described by ordinary or partial differential equations whose solutions are given by holomorphic or meromorphic functions in the complex domain. In many cases, only the magnitude of these functions are observed on various points on the purely imaginary $j\omega$-axis since coherent measurement of their phases is often expensive. However, it is desirable to retrieve the lost phases from the magnitudes when possible. To this end, we propose a physics-infused deep neural network based on the Blaschke products for phase retrieval. Inspired by the Helson and Sarason Theorem, we recover coefficients of a rational function of Blaschke products using a Blaschke Product Neural Network (BPNN), based upon the magnitude observations as input. The resulting rational function is then used for phase retrieval. We compare the BPNN to conventional deep neural networks (NNs) on several phase retrieval problems, comprising both synthetic and contemporary real-world problems (e.g., metamaterials for which data collection requires substantial expertise and is time consuming). On each phase retrieval problem, we compare against a population of conventional NNs of varying size and hyperparameter settings. Even without any hyper-parameter search, we find that BPNNs consistently outperform the population of optimized NNs in scarce data scenarios, and do so despite being much smaller models. The results can in turn be applied to calculate the refractive index of metamaterials, which is an important problem in emerging areas of material science.

## 1    Introduction

Numerous physical systems are described by ordinary or partial differential equations whose solutions are given by holomorphic or meromorphic functions $f(z) = |f(z)| \exp(j\theta(z))$ in the complex domain, where $|f(z)|$ and $\theta(z)$ are respectively the magnitude and phase of $f(z)$. In many cases, only $|f(j\omega)|$, the magnitude of $f(z)$, is observed on a finite set of points ($\Omega$) of the purely imaginary $j\omega$-axis[1], since the coherent measurement of their phases may be expensive (see Ch. 13 of Stern (2016)) or require significant expertise (King, 2009). However, it is desirable to retrieve the lost phase $\theta(j\omega)$ from measurements of the magnitudes $|f(j\omega)|$ for $\omega \in \Omega$, since the phase often encodes important information. For example in both spectroscopy and optical imaging, estimation of $\theta(j\omega)$ is often not possible. However, phase information determines the optical constants in the spectroscopy of materials (Lucarini, 2005), and is more important than $|f(j\omega)|$ in optical imaging (Stern, 2016). The phase retrieval problem is also important in a number of other disciplines, including holography (Fienup, 1982), x-ray crystallography (Nugent, 2007), astronomy (Stark, 1987), blind deconvolution (Shechtman et al., 2015), and coherent diffractive imaging (Latychevskaia, 2018).

---

[1]For readers unfamiliar with phase retrieval problems and have questions about why function values are only observed on the purely imaginary line, please refer to the explanation we have included in the Appendix.

Many techniques for phase retrieval have been developed in the literature. The Kramers–Kronig formula, see Lucarini (2005) (also known as Sokhotski–Plemelj formula), is noteworthy and is derived by the application of the Hilbert Transform to $\log f(j\omega) = \log |f(j\omega)| + j\theta(j\omega)$. Unfortunately, for phase retrieval, this elegant formula requires knowledge of the magnitude $|f(j\omega)|$ along the entire $j\omega$-axis. However, as mentioned above, acquiring these values is often impossible since $\Omega$ is typically a finite set with a small number of elements. This severely limits the application of Kramers–Kronig's formula in many applications. Another approach is based on compressed sensing, but it requires sparsity assumptions, and stringent assumptions on the design or projection matrix (i.e., Restricted Isometry Property (RIP) (Candes, 2008), mutual incoherence (Foucart & Rauhut, 2013), etc) that often do not hold for meromorphic functions encountered in practice.

To overcome the above limitations, phase retrieval can also be cast as a data-driven regression problem, where we wish to learn a mapping from the magnitude of a signal, $X \in R^N$, to its phase, $\Theta \in \mathbb{C}^N$. If the number of frequencies ($|\Omega|$) measured is $N$ then the phase can be represented as a vector of complex numbers $\Theta = [\exp(\theta(j\omega_1)), \cdots, \exp(\theta(j\omega_N))] \in \mathbb{C}^N$, where $\exp(\theta(j\omega_i))$ is the phase at $i$-th frequency. Similarly, the signal's corresponding magnitude vector can be given by $|f(j\omega_i)|, i = 1, 2, \cdots, N$, and the goal is to infer $\Theta$ based upon $X$. We can then learn this mapping using a data-driven model (see Figure 1 below for an example).

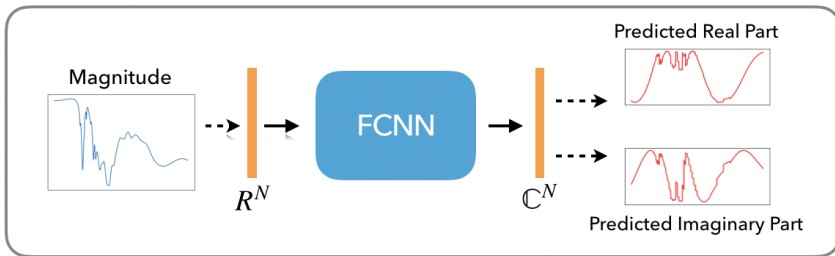

Figure 1: Example of the deep neural network approach to phase retrieval problem. FCNN refers fully-connected neural network here but can be replaced by any machine learning model. The magnitude is encoded into an input of size $N$ while the output is an $N$-dimensional complex vector of corresponding phases.

With a sufficient amount of training data, it is likely that a universal function approximator (i.e. neural network), can learn the underlying function well within a tolerance range. For instance, deep learning has shown great success on phase retrieval problems in the imaging area where abundant training data is available (Sinha et al., 2017; Metzler et al., 2018). But sufficient training data can be expensive to acquire in many practical applications, or sometimes even impossible.

In this paper, we depart from the existing literature by exploring the applications of physics-infused neural networks to this setting. The infusion of physics reduces the number of required training examples to a small enough amount that is practical for many measurements and design applications of interest. The main idea of the approach is described next. Given the fact that any phase value has a magnitude of 1 (i.e. $|\exp(j\theta(j\omega))| = 1$ for all $\omega$), we observe that conditions of a Theorem of Helson and Sarason are then satisfied indicating that $\exp(j\theta(j\omega))$ can be approximated by a rational function whose numerator and denominator are both Blaschke products (Garcia et al., 2016). This is then used to design a neural network for calculating coefficients of these Blaschke products which may be be used for phase retrieval.

In Section 2, we briefly review conventional phase retrieval methods, and existing applications of the neural network to phase retrieval problems. In Section 3, we briefly review Blaschke products and the Helson-Sarason Theorem. In Section 4, we propose our approach for baking these results into neural networks resulting in Blaschke Product Neural Networks. In Section 5, we provide experimental results on various synthetic and metamaterials datasets along with a inclusive set of baselines to demonstrate the performance improvements of our construction.

## 2 RELATED WORK

We first briefly review existing methods of phase retrieval applicable to phase retrieval of analytic and meromorphic functions of interest.

Perhaps the most important result in this area is the elegant calculation of Kramers and Kronig (also known as Sokhotski–Plemelj formula) obtained by the application of the Hilbert Transform to $\log f(j\omega) = \log |f(j\omega)| + j\theta(j\omega)$ and is given by

$$\theta(j\omega) = \frac{2\omega}{\pi} P.V. \int_0^\infty \frac{\log |f(j\beta)|}{\beta^2 - \omega^2} d\beta,$$

where P.V. denotes Cauchy's principal value. We note that this can also be derived by invoking the Cauchy integration formula along a path containing the imaginary $j\omega$-axis.

However, there is a major obstacle in the application of this elegant formula. In many important practical scenarios, the set $\Omega$ of frequencies (points on $j\omega$-axis) that we have measurements of $|f(j\omega)|$ has a small cardinal number. In such cases, the approximation of Kramers-Kronig integral by a Riemann Sum produces unsatisfactory results. Other approaches used to analytically extend or extrapolate $|f(j\omega)|$ have also demonstrated limited success (Chalashkanov et al., 2012).

Another approach to phase retrieval is by sparse representation. This approach assumes that it is a priori known that $f(z)$ has a sparse representation in a known dictionary of functions $\mathbb{D} = \{\Phi_i(z), i = 1, 2, \cdots, M\}$ i.e. it can be written as a linear sum of elements of $\mathbb{D}$ with only a small number of non-zero coefficients. Under this assumption, many techniques are developed to recover these coefficients and the phase of $f(z)$. Again, this is an interesting proposal but the underlying assumptions may not hold in many problems of interest. In particular, neither a dictionary $\mathbb{D}$ may be known nor classes of functions of interest may be sparse in any fixed dictionary. Recently, some new techniques based on deep learning have been developed to learn the structure of the underlying signal which we want to recover its phase. That is, deep neural networks are used to encode the prior knowledge on the structure of the signal instead of hard-coded assumptions such as sparsity (Jagatap & Hegde, 2019). However, these methods require a significant number of training samples from the magnitude of the underlying signal, which limits the applicability of these approaches on material science applications.

Recently, DL has demonstrated its capability to solve the phase-retrieval problem, especially in the imaging area: Sinha et al. (2017) achieved lensless imaging for phase objects utilizing CNN. Xue et al. (2019) constructed a bayesian convolution neural network to achieve phase imaging while quantifying the uncertainty of the network output. Metzler et al. (2018) combined regularization-by-denoising framework with a convolution neural network for noisy phase retrieval for images. Goy et al. (2018) achieved low photon count phase retrieval with a Convolutional Neural Network(CNN) as well. To avoid the difficulty of getting a large set of paired phase and intensity measurements, Zhang et al. Zhang et al. (2021) used an unpaired dataset for phase-retrieval. Other works of Tayal et al. (2020); Houhou et al. (2020) have also used deep learning for end-to-end phase recovery. None of these works use the underlying physics in their deep learning approach or they require a large number of training samples; hence, are not directly applicable to the physics applications we are interested in this paper. We took a different path of applying physical prior knowledge to our phase retrieval process.

One closely related line of work to ours is solving partial differential equations using neural networks (Dissanayake & Phan-Thien, 1994; Berg & Nyström, 2018; Zhang & Bilige, 2019; Chen et al., 2018). However, these methods typically need a lot of data and require careful selection of hyper-parameters; additionally, they do not consider directly solving the phase retrieval problem. Our approach, on the other hand, needs no specific hyper-parameter search algorithm, and achieves high-accuracy result with a very limited training samples.

## 3 MATHEMATICAL BACKGROUND

We briefly review the mathematical background required in the rest of this paper. In particular, we review Blaschke products and the Helson-Sarason Theorem. We note that the standard presentation of the Blaschke products is motivated by inner functions in Hardy spaces presented over the unit disk

whose boundary is the unit circle (Rudin, 1987; Garcia et al., 2016). In our case, we are interested in the right hand half plane whose boundary is the $j\omega$-axis. These two are equivalent using the complex Mobius transformation $(z+1)/(1-z)$. In this light, our exposition is closer to Akutowicz (1956).

Let $a_1, a_2, a_3, \cdots$ be an infinite sequence of numbers for which $\Re(a_k) \geq 0$ for all $k$ for which

$$\sum_{j=1}^{\infty} \frac{\Re(a_j)}{|a_j|^2 + 1} < \infty,$$

then for any integer $n$, any complex function given by

$$B(z) = \left(\frac{1+z}{1-z}\right)^n \prod_j \frac{|a_j - 1|}{a_j - 1} \frac{|a_j + 1|}{a_j + 1} \frac{z - a_j}{z + \bar{a}_j}$$

is referred to as a infinite Blaschke Product. If the set $\{a_1, a_2, \cdots, a_k\}$ is finite, $B(z)$ is referred to as a finite Blaschke product. For a finite Blaschke Product, it is clear that $\sum_{j=1}^{k} \frac{\Re(a_j)}{|a_j|^2+1} < \infty$ since there are only finite number of coefficients $a_j$, and therefore this condition can be dropped. Note that roots with $\Re(a_k) = 0$ do not contribute to the product.

For both finite and infinite Blaschke products, we can see that $|B(z)| = 1$ for $z = j\omega$ on the purely imaginary axis [2]. In this paper, we are only interested in finite Blaschke products because of the following approximation Theorem of Helson and Sarason.

**Theorem 1** *Let $u(j\omega)$ be any continuous function on the $j\omega$-axis for which $|u(j\omega)| = 1$ for all $\omega$. Let $\epsilon > 0$. Then there exists finite Blaschke products $B_1(j\omega)$ and $B_2(j\omega)$ such that*

$$\|u(j\omega) - \frac{B_1(j\omega)}{B_2(j\omega)}\|_\infty < \epsilon,$$

*where the $\|\cdot\|_\infty$ denotes the standard $L^\infty$ norm computed over the $j\omega$ axis.*

Let $b(j\omega) = \exp(j\theta(j\omega))$. We know that $|b(j\omega)| = 1$ because $|b(j\omega)| = |\exp(j\theta(j\omega))| = |\cos(\theta(j\omega)) + j\sin(\theta(j\omega))| = \sqrt{\cos(\theta(j\omega))^2 + \sin(\theta(j\omega))^2} = 1$. Applying the Helson-Sarason Theorem, for analytic or meromorphic functions $f(j\omega)$ (with no poles on the imaginary axis), the phase $b(j\omega)$ can be approximated by a rational function of finite Blaschke products. *The main theme of this paper is that these Blaschke products can be in turn computed using neural networks with a small amount of data.*

From the expression of finite Blaschke products, we can see that $B_i(j\omega) = A_i P_i(j\omega)/P_i^*(j\omega)$ for $i = 1, 2$ where $A_i$ a complex number with $|A_i| = 1$ and $P_i(j\omega)$ is a complex polynomial of $\omega$ and $P_i^*(z) = \bar{P}_i(-\bar{z})$, where $\bar{P}_i(j\omega)$ is constructed from $P_i(j\omega)$ by replacing all its coefficients with their respective conjugates. Note that $[P^*(j\omega)]^* = P(j\omega)$. Let $A = A_1/A_2$ and represent the Blaschke products $B_1$ and $B_2$ in the above theorem with the complex polynomials, we have the approximating rational function written as $AP_1(j\omega)P_2^*(j\omega)/P_1^*(j\omega)P_2(j\omega)$ with $|A| = 1$. Thus we have the estimation for phase function $b(jw)$ as:

$$b(jw) \approx A \frac{P_1(jw)P_2^*(jw)}{P_1^*(jw)P_2(jw)} = A \frac{P(jw)}{P^*(jw)}, \tag{1}$$

where $P(z) = P_1(z)P_2^*(z)$ and $P_1(z)$ and $P_2^*(z)$ correspond to respectively factors of $P(z)$ with roots with positive and negative real parts.

## 3.1 Blaschke Product Neural Network

Inspired by the above, we model the phase function by equation 1, and propose the Blaschke Product Neural Network (BPNN). A BPNN is a neural network with the magnitude of observations as the input and the coefficients of the rational function of Blaschke products as the output. In this paper, we only present BPNNs which are fully connected neural networks (FCNNs), although such a restriction is not necessary. We next describe the operation of BPNNs by discussing the forward and backward pass mechanisms.

---

[2]Details proving $|B(jw)| = 1$ can be found in the appendix

**Forward Pass**    The output of BPNN are Blaschke coefficients of the complex polynomials that are in turn used to construct predicted phase function $\tilde{b}(j\omega)$ according to equation 1. Without loss of generality, let $\Omega = \{\omega_1, \omega_2, \cdots, \omega_N\}$ be the set of frequencies for which experimental results measuring the magnitude of signals $|f(j\omega)|$ is performed, and their corresponding phases must be retrieved. It can be assumed that $\omega_0 < \omega_1 < \cdots < \omega_N$. The BPNN then produces estimates $\tilde{b}(j\omega_i)$ of the phase $b(j\omega_i)$. Combining with the known magnitude at $\omega_i$, this gives an estimate

$$\tilde{f}(j\omega_i) = |f(j\omega_i)|\tilde{b}(j\omega_i).$$

**Backward Pass**    After predictions are generated with forward pass , the prediction error is computed as

$$L = \frac{1}{N}\sum_{i=1}^{N}|f(j\omega_i) - \tilde{f}(j\omega_i)|^2.$$

The training seeks to minimize $L$ over all training samples in a standard manner using the stochastic gradient descent algorithm and its variants. Cross-validation and testing can also be done in a standard format.

An important hyper-parameter of interest is number of degrees for complex polynomials $P_1(j\omega)$ and $P_2(j\omega)$. This hyper-parameter can either be cross-validated or chosen based on the practitioner's knowledge about the complexity of the specific phase retrieval problem. In our experiments, we found that complex phase retrieval problems may require larger degree polynomials, and calculating the value of these high degree polynomials over large frequency ranges (100THz - 1500THz and above) may lead to the overflow problem. To overcome this problem, we propose a piece-wise implementation of BPNN.

**Piece-wise Implementation of BPNN**    In this approach, given a large frequency range, the whole frequency band is partitioned into non-overlapping contiguous segments. The Blaschke method is then applied to each segment (see figure2b). The BPNN is then trained to output the respective Blaschke coefficients for each segment. During training, the predictions of phases on all segments are concatenated. Then the following training is the same as non-piecewise BPNN. The exact method of partitioning and degrees of corresponding Blaschke polynomials for each sequence are hyper-parameters of choice. We have two example applications of piece-wise BPNNs to metamaterial datasets in our numerical experiments.

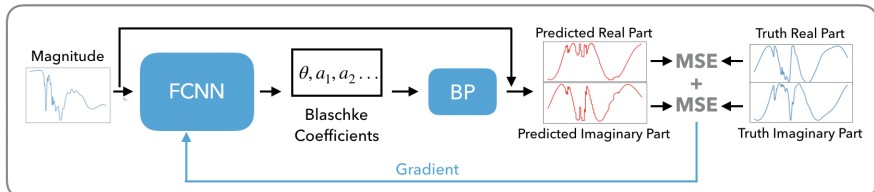

(a) Illustration of BPNNs

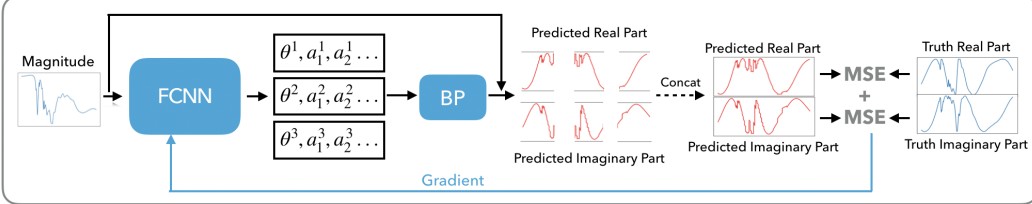

(b) Illustration of Piece-wise Implementation of BPNN

Figure 2: Structure of BPNN (and piecewise BPNN): BPNN is an end-to-end model. Predictions are constructed by the combination of the Blaschke coefficients output of FCNN and values of input magnitude. In the piece-wise BPNN case, FCNN generates the sets of Blaschke coefficients for all segments (partitions) of the frequency range, and predictions are made for each segment separately and the results are concatenated.

## 4 EXPERIMENTS

We conduct simulations on five phase retrieval datasets, two of which are synthetic, while the remaining three are metamaterial datasets. A detailed listing of the datasets is shown in Table 1. We endeavor to test the BPNN on diverse, yet complete, datasets, and thus we choose datasets that include both holomorphic and meromorphic complex functions. Additionally, the selected metamaterial datasets are comprehensive, and well represent the discipline.

### 4.1 DATASETS

Table 1: List of phase retrieval problems in Experiments section. (# Frequency) shows total number of frequency points considered for the problem. Please see relevant sections for detailed descriptions.

| PHASE RETRIEVAL PROBLEM | TYPE | # FREQUENCY |
|---|---|---|
| Polynomial ODE (4.1.1) | Synthetic | 200 |
| Lorentzian response function (A.7) | Synthetic | 1000 |
| Metamaterial absorber (4.1.3) | Metamaterial | 1000 |
| Huygens' metasurface (4.1.4) | Metamaterial | 2001 |
| Membrane metasurface (4.1.5) | Metamaterial | 2001 |

#### 4.1.1 SYNTHETIC I: POLYNOMIAL ODE

We first select points $\omega_0 < \omega_1 < \omega_2 < \cdots < \omega_N$ on the $j\omega$ axis and fix these values. The first synthetic dataset is the solution to a ordinary differential equation (ODE) given by:

$$\frac{df}{dz} = p(z),$$

where $p(z)$ is a complex polynomial of degree $m = 4$ with its roots lying on the unit circle. We uniformly sample roots of $p(z)$ on the unit circle, then solve the above ODE on the $j\omega$-axis using the Euler method with the real and imaginary part of the initial condition $f(\omega_0)$ uniformly selected in $[-1, 1]$ for 1050 randomly generated initial values. The values $(f(j\omega_1), f(j\omega_2), \cdots, f(j\omega_n))$ are then calculated and their magnitudes and phases are used in this experiment, of which 50 are used for training and 1000 for testing.

#### 4.1.2 SYNTHETIC II: LORENTZIAN RESPONSE FUNCTION

The second synthetic dataset is generated by considering a causal model of a non-magnetic material, with a frequency-dependent complex dielectric permittivity defined as a sum of Lorentzian oscillators. A simple Lorentzian model of the dielectric function is applicable to a wide range of resonant material systems with response functions that must obey Kramers-Kronig relations. Using the transfer-matrix method, Markoš & Soukoulis (2008), the complex transmission coefficient $t(\omega)$ can be computed directly from the dielectric function via closed form equations.(Please see appendix for more details). We generate a dataset of complex transmission coefficients $t$ by uniformly sampling Lorentzian parameters for the frequency range 100-500 THz. We consider a fixed set of four Lorentzian oscillators, and generate 50 sample transmission spectra for training, and 2000 spectra for the test dataset.

#### 4.1.3 METAMATERIAL I: METAMATERIAL ABSORBER

Metamaterials are structured materials consisting of periodic arrays of resonant elements that derive their spectral properties from their unit cell geometry Smith et al. (2000). With effective material properties that can consist of resonances in both the electric permittivity and magnetic permeability, and hence more unknown material parameters, metamaterials often represent a more complicated phase-retrieval problem than many conventional systems.

The first metamaterial dataset considers a metal-based absorber geometry with a metallic ground plane backing a dielectric spacer layer, designed to operate in the millimeter wave (50-200 GHz) frequency range. This metamaterial absorber (MMA) dataset has 55,000 pairs of MMA geometry and complex frequency dependent reflection coefficient $r(\omega)$ ($t(\omega)$ is zero due to the metal backing). The data are obtained with commercial computational electromagnetic simulation software (CST Microwave Studios).

### 4.1.4 METAMATERIAL II: HUYGENS' METASURFACE

Huygens' metasurfaces made of dielectric materials represent another class of metamaterials Decker et al. (2015), with underlying properties that are different than their metallic counterparts, including for example the lack of Ohmic losses Liu et al. (2017). When individual designs are combined into supercells, all-dielectric metasurfaces can exhibit very rich spectral phenomena. The Huygens' metasurface dataset consists of the complex reflection and transmission response derived from a supercell of elliptical resonators adopted from the work of Deng et al. (2021). The phase information can be inferred directly from the complex reflection or transmission. Out of a total of 1000 samples, we randomly sample 80 for training and reserve the rest for testing.

### 4.1.5 METAMATERIAL III: MEMBRANE METASURFACE

The membrane metasurfaces are similar to the Huygens' metasurfaces. However, instead of having four stand-alone elliptical resonators in the supercell, it consists of a thin slab of SiC with regions of empty holes. The SiC slab filling the supercell has a thickness defined by $h$, and a periodicity $p$. In the SiC slab, we placed four elliptical holes where the elliptical SiC resonators were placed. The four holes thus have the same geometrical parameters $r_{ma}$, $r_{mi}$, and $\theta$. The fourteen geometrical parameters share the same range as the elliptical resonators dataset. Similarly, the input of these datasets contains fourteen geometry parameters, and the outputs are complex reflection and transmission with 2001 frequency points range from 100 THz to 500 THz. Out of a total of 7331 samples, we randomly sample 80 for training and reserve the rest for testing.

## 4.2 BENCHMARKING NEURAL NETWORKS

Comparison is done on each dataset with training datasets of increasing size. For each phase retrieval problem and each training dataset, we compare BPNN to a family of NNs of various sizes and structures. For each structure, we search its hyper-parameters of training to guarantee we have NNs close to optimal. And we record test error at each training epoch and report the best test error. We note that recording test error is uncommon but this guarantees that the reported performance is close to its limit.

We use NNs of various structures and all hidden layers of each of the NNs have the same width. The range of hidden layer width in search is [32,64,128,256] and the number of hidden layers in search is [1,2,3,4]. For hyper-parameters of training, we search dropout rate in the list of [0., 0.05, 0.1, 0.15, 0.2] and learning rate in the list of [1e-4, 5e-4, 1e-3, 5e-3, 1e-2]. Each NN is trained with 6000 epochs after which training losses of all neural networks have converged. The training is performed with 3 random initialization and we report the best test error. Huygens' metasurface dataset is more complex than the other datasets, so the range of hidden layer size in search is increased to [64,128,256,512] while keeping the ranges of the other searched parameters the same. Unlike the extensive searching for NNs, we do not search hyper-parameters of BPNN and only use one BPNN for each phase retrieval problem. We also train BPNN with 3 random initialization on each dataset and report its best test error. We use ReLU as activation function for both NNs and BPNNs while Adam(Kingma & Ba, 2017) is used for training of all NNs and BPNNs.

## 4.3 RESULTS

For each dataset, we compare the family of regular fully-connected neural networks, each with its best searched hyper-parameters to BPNN without hyper-parameters search. For three meta-material datasets, we further include four baselines: (1)The Kramers-Kronig method (KK-method): a conventional phase retrieval method (Lucarini, 2005), (2)AAA-network algorithm: a established rational approximation technique, similar to BPNN we used its functional form to approximate the phase

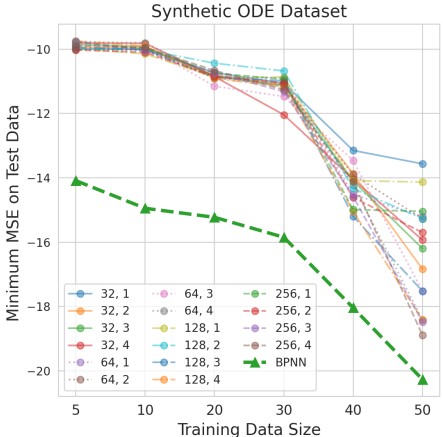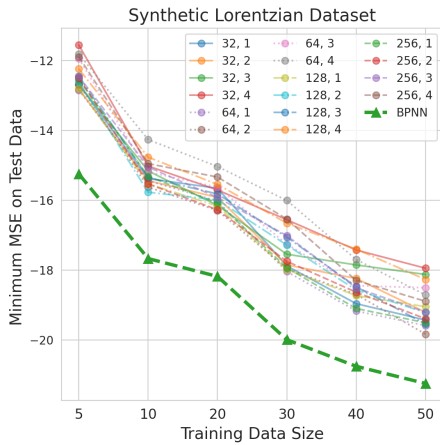

Figure 3: Performance comparison of a number of neural networks of different structures, each with its best hyper-parameters, and BPNN on two synthetic phase retrieval problems with training set of various size.

function (Nakatsukasa et al., 2018b), (3)MUSIC algorithm: a data-driven function approximation method that estimates functions as sum of exponentials (Schmidt, 1986a), and (4) Linear+BP: using linear regressor instead of neural networks in BPNN.(Please see Appendix for more details about KK-method, AAA algorithm and MUSIC algorithm.) All y-axes are plotted with a dB scale in base 10 ($10 \log_{10}(e)$ where $e$ is the MSE error) for visual clarity. For the legends in Fig. 3 we use entries of the form (A, B), where A refers to the width (i.e., number of neurons) in each hidden layer of the neural network being evaluated, and B refers to the depth of the neural network (i.e., number of hidden layers).

**Performance of Baselines** We note that both the KK and MUSIC methods exhibit poor performance, because the KK method requires a large number of magnitude measurements to achieve accurate estimates, while the MUSIC method expects that the target function is sparse in some basis, which cannot be assumed for our problems. The Linear+BP model also performs substantially worse than the the BPNN, suggesting that the relationship between the phase magnitude an the Blaschke Product coefficients is non-linear. Although representing the phase function with a Blaschke Product decreases the complexity of the problem, mapping from magnitude to the Blaschke coefficients still appears to be beyond the capability of a linear regressor. In the following paragraphs we describe results of each dataset in detail.

**Synthetic Datasets** We use BPNN of 4 roots with a fully-connected neural network of 2 hidden layers, each of size 64 and no dropout. It is obvious from figure[3] that BPNN outperforms a family of optimized neural networks by a large margin. This supports our theoretical justification of using BPNN when the phase functions are precisely meromorphic functions. The outstanding performance of BPNN on the Lorentzian dataset implies its value on phase retrieval problems in metamaterials. To confirm this, we conduct more experiments on real metamaterial datasets.

**MMA, ADM, Membrane** For the MMA dataset, similar to synthetic datasets, we also use BPNN with 4 roots while using a fully-connected neural network with 2 hidden layers of size 64. No dropout nor weight-decay was added. Although without any regularization techniques for neural networks, BPNN still demonstrate superior performance to all the optimized neural networks. Given that ADM dataset and Membrane dataset are much more complex than the synthetic datasets and MMA dataset, we use a Piecewise BPNN with frequencies partitioned into 20 equal-length sequences and we assign each sequence with 3 roots. We note that this is the most naive split for Piecewise BPNN and a more reasonable split can lead to improved performance. We find that the best errors of regular neural networks are close to BPNN's best error. However, we also find that the median error of BPNN is much smaller than the median error of the best neural network (the neural network with smallest test MSE error across all architectures). This shows that the BPNN is making continued good predictions through the sacrifice of small amounts of poor predictions. This property can be valuable when the prediction problem at hand is the on-or-off type due to BPNN's higher probability of making good predictions.

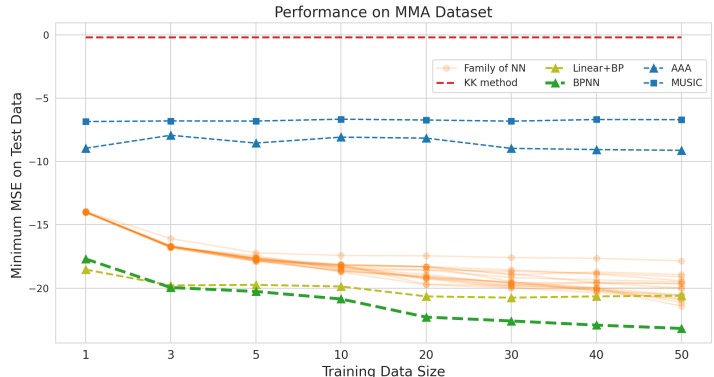

Figure 4: Performance comparison of family of neural networks, each with its best hyper-parameters and BPNN (without hyper-parameter search) on MMA dataset along with four other baselines: the KK method, linear regressor+Blaschke Product, MUSIC algorithm and the AAA method.

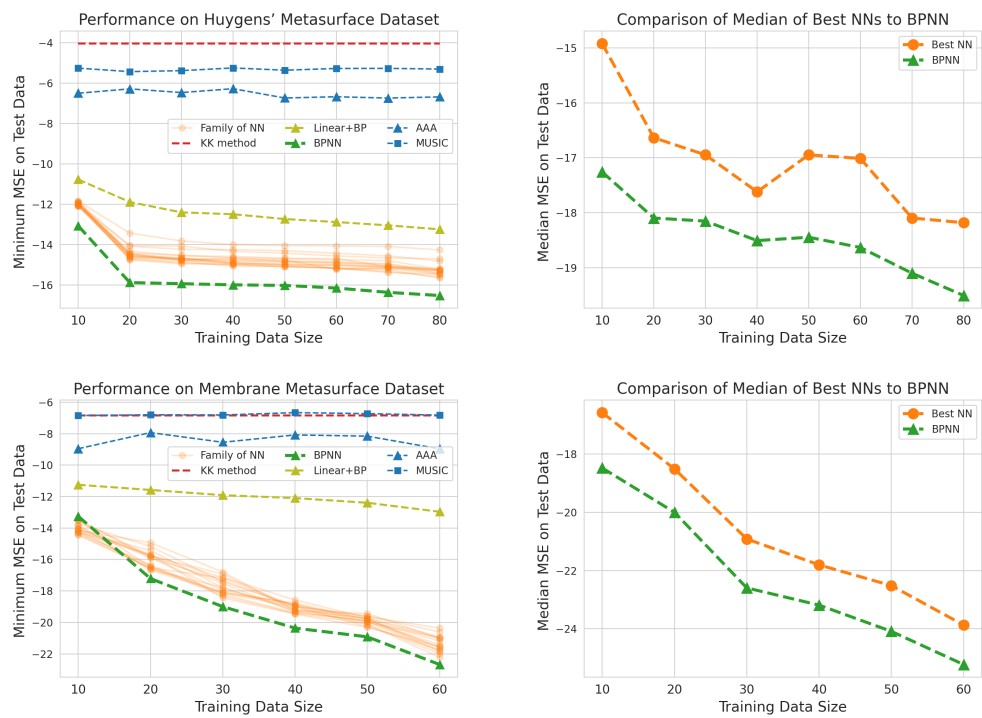

Figure 5: On the left is the performance comparison of family of neural networks, each with its best hyper-parameters and BPNN along with four other baseline methods. On the right is the comparison of median MSE loss of the best neural network in the large family and median MSE loss of BPNN. (On top are performance comparisons on Huygens' metasurface dataset; At bottom are performance comparisons on membrane metasurface dataset.)

## 5 CONCLUSION

We presented a physics-infused neural network referred to as Blaschke Product Neural Network (BPNN) for phase retrieval problem of holomorphic and meromorphic functions. Through extensive experiments on synthetic and metamaterial datasets, we have found that the data required for training neural networks is significantly less than expected when the underlying functions are holomorphic or meromorphic. By baking the physics into neural networks, even less training data is required and the performance is further improved.

## 6 REPRODUCIBILITY STATEMENT

We upload four datasets. Due to restriction of maximum supplementary size of 100MB, we include three datasets in supplementary: two synthetic datasets and Huygens' metasurface dataset. Membrane metasurface can be accessed with this link (https://figshare.com/s/966f196d1847269a447c). Details about hyper-parameter search space and the process of searching can be found in the experiment section. Included in the supplementary we also have our implementation of the BPNN and neural networks by Pytorch with easy option of the hyper-parameters.

## ACKNOWLEDGMENTS

This work was supported in part by the Office of Naval Research (ONR) under grant number N00014-21-1-2590. YD, OK, SR, JM, and WJP acknowledge funding from the Department of Energy under U.S. Department of Energy (DOE) (DESC0014372).

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

# A APPENDIX

## A.1 OBSERVATIONS OF PHASE FUNCTION VALUES

Here we provide a short explanation for the reason why the function values are only available on the imaginary line. In physics, electrical engineering and other fields, solutions to many linear PDEs with constant coefficients, and also many other measurements are done in the frequency domain. This in principle is achieved by multiplying a real signal x(t) by sinusoidals $\cos(\omega t)$ and $\sin(\omega t)$, and integrating over time (Calculating Cosine and Sine Transforms) for various values of $\omega$. If x(t) is a real physical signal, these cosine and sine transforms are real and imaginary parts of $X(j\omega)$, the complex Fourier transform of x(t). Thus $X(j\omega)$ gives the value of $x(\cdot)$ over the imaginary axis.

## A.2 $|B(jw)| = 1$

We provide the details proving the norm of Blashke product is 1 if $z = jw$. To show that it has a norm of one, we should show that all component (that are multiplied) would have a norm of one individually:

$$B(z) = \left(\frac{1+z}{1-z}\right)^n \prod_j \frac{|a_j - 1|}{a_j - 1} \frac{|a_j + 1|}{a_j + 1} \frac{z - a_j}{z + \bar{a}_j}$$

$$B(z) = B_1(z) * B_2(z) * B_3(z)$$

$$|B(z)| = |B_1(z)| * |B_2(z)| * |B_3(z)|$$

$$B_1(z) = \left(\frac{1+z}{1-z}\right)^n ; B_2(z) = \prod_j \frac{|a_j - 1|}{a_j - 1} \frac{|a_j + 1|}{a_j + 1} ; B_3(z) = \prod_j \frac{z - a_j}{z + \bar{a}_j}$$

$$|B_1(jw)| = |\left(\frac{1+jw}{1-jw}\right)^n|$$

$$= |\left(\frac{1+jw}{1-jw}\right)|^n$$

$$= \left(\frac{|1+jw|}{|1-jw|}\right)^n$$

$$= \left(\frac{(1+w^2)^{0.5}}{(1+w^2)^{0.5}}\right)^n$$

$$= 1$$

$$|B_2(jw)| = |\prod_j \frac{|a_j - 1|}{a_j - 1} \frac{|a_j + 1|}{a_j + 1}|$$

$$= \prod_j \frac{||a_j - 1||}{|a_j - 1|} \frac{||a_j + 1||}{|a_j + 1|}$$

$$= \prod_j 1 * 1$$

$$= 1$$

$$|B_3(jw)| = |\prod_j \frac{z - a_j}{z + \bar{a}_j}|$$

$$= \prod_j \frac{|jw - a_j|}{|jw + \bar{a}_j|}$$

$$= \prod_j \frac{|jw - a_j - b_j j|}{|jw + a_j - b_j j|} (\text{assume } a_j = a_j + b_j j)$$

$$= \prod_j \frac{(a_j^2 + (b_j - w)^2)^{0.5}}{(a_j^2 + (w - b_j)^2)^{0.5}}$$

$$= 1$$

### A.3 Hyper-parameters of Piecewise BPNN

In order to provide analysis and some insights on hyper-parameters of Piecewise BPNN, we conduct experiments on the two most complex datasets, Hyugens' metasurface dataset and membrane metasurface dataset, to have empirical results regarding the effect of hyper-parameters (e.g number of segments and number of roots per segment) on performance. We use training dataset of size 50 for both datasets and fix BPNN's neural networks to have 2 hidden layers and hidden size of 64 for each layer. From figure[6] we can see that Piecewise BPNN's performance is robust to hyper-parameters: a wide range of different settings of hyper-parameters can generate satisfactory results as long as the total number of roots is above the threshold necessary. Similarly, the Piecewise BPNN is also robust to overfitting caused by high number of roots: while the optimal total number of roots appears in the range of 60-80, overshooting total number of roots to 160 or even 200 doesn't cause a sharp increase in test error. We believe this is a strong indication that the Piecewise BPNN learns the necessary complexity and thus zeros out the unnecessary degrees

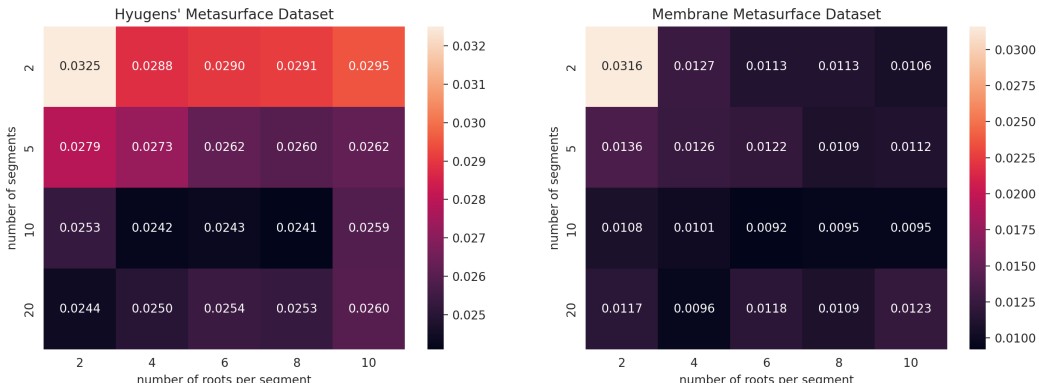

Figure 6: On the left is the heatmap of minimum test MSE (mean squared error) of Piecewise BPNN with different hyper-parameter configurations on Hyugen's metasurface dataset. On the right is the heatmap of minimum test MSE of Piecewise BPNN with different hyper-parameter configurations on membrane metasurface dataset. Each minimum MSE is calculated with 3 runs of training from different initialization.

### A.4 Baseline model 1: Kramers–Kronig

The Kramers–Kronig relations, implied by causality of a stable physics system, are often used to convert between imaginary parts and real parts of physical functions such as dielectric functions, susceptibility, and reflection coefficient. We adopt the notation from Grosse & Offermann (1991), since the metamaterials problems included in the main text focus on the phase retrieval tasks on spectral data, including reflectance. In general, the Kramers–Kronig relations for complex reflection coefficient are given by:

$$r'(\omega) = \frac{1}{\pi} P \int_{\infty}^{-\infty} \frac{r''(\omega')}{\omega' - \omega} d\omega'$$

$$r''(\omega) = -\frac{1}{\pi} P \int_{\infty}^{-\infty} \frac{r'(\omega')}{\omega' - \omega} d\omega'$$

which $r'(\omega)$ and $r''(\omega)$ are real and complex part of reflection coefficient $r(\omega)$ respectively.

In practice, $r'(\omega)$ and $r''(\omega)$ are often unknown. The reflectance $R(\omega) = |r(\omega)|^2$ is the physical quantity that is directly measurable by intensity measurement, but the phase information is lost.

Therefore, it is beneficial to derive phase $\phi(\omega)$ from reflectance $R(\omega)$. A general form of complex reflection coefficient can be written as $r(\omega) = \sqrt{R(\omega)} \exp i\phi(\omega)$. We can infer that:

$$\ln r(\omega) = 0.5 \ln R(\omega) + i\phi(\omega)$$

If Kramers-Kronig relations are valid for $\ln r(\omega)$, we can first measure $R(\omega)$ from zero frequency to the highest measurable frequency. Then we can apply Kramers-Kronig relations on $0.5 \ln R(\omega)$ as real part of $\ln r(\omega)$ to infer the imaginary part $\phi(\omega)$. In our implementation, we follow the above procedure to derive $\ln r(\omega)$, and therefore can get the Kramers-Kronig reconstructed complex reflection coefficient $\hat{r}(\omega)$. Since the loss metric in the main text is the averaged mean square error of the real and imaginary parts of the reflection coefficient, $\hat{r}'(\omega)$ and $\hat{r}''(\omega)$ are further taken from $\hat{r}(\omega)$ to be compared with $r'(\omega)$ and $r''(\omega)$.

The above procedure is built on the premise that there is causality for $\ln r(\omega)$, but $\ln r(\omega)$ is not a real physical function that is proven to be governed by causality. Thus, we cannot prove that Kramers-Kronig relations is valid on $\ln r(\omega)$. Furthermore, the reflection spectrum in the metamaterials dataset are typically in a finite frequency range that degrades the performance of the Kramers-Kronig relations. Nonetheless, we provided the Kramers-Kronig relations as a baseline to perform phase retrieval.

### A.5 BASELINE MODEL 2: AAA-NETWORK ALGORITHM DETAILS

Here we supply the details of the Adaptive Antoulas-Anderson (AAA) -network baseline we included in our main text. Following the notation of Nakatsukasa et al. (2018a), it uses rational barycentric representation of rational apprixmation, which approximate an arbitrary function as rational function of type (m, m):

$$r(z) = \frac{n(z)}{d(z)} = \sum_{j=1}^{m} \frac{w_j f_j}{z - z_j} \bigg/ \sum_{j=1}^{m} \frac{w_j}{z - z_j}$$

where $z_j, w_j, f_j$ are support points, weights and value parameters of this approximation. It approximates a complex function over a set of complex support points using an iterative greedy selection and uses linear least-squares to solve for weights.

Although AAA also utilizes rational approximation, there are a couple of fundamental differences between AAA and our Blashke product approximation. (1) The functional form difference: AAA uses the barycentric representation, which although limited to type (m, m) rational, covers a much larger functional space compared to our Blashke product, that is specialized to approximate the phase function on the unit circle according to the Helson-Sarason Theorem. (2) The fitting process: AAA uses a greedy solution to pick the support point first and then solve the least square for weights and values. Our Blashke product fits $a_j$ simultaneously and does not require iterative steps internally (although the actual fitting is done using gradient-based methods, which requires iterative steps). This means the AAA is difficult to build into a neural network and get gradient feedback from the fit, unlike our BPNN approach.

With AAA, we can successfully approximate the phase function using three sets of barycentric parameters (also known as AAA parameters). However, this process requires knowledge of the phase function or at least the ability to sample points on the phase function. Therefore a naive AAA alone is not able to be applied to our application of phase retrieval, which not only requires the approximation of phase function but also the extrapolation of the mapping between magnitude and the approximation parameters.

For programming implementation, we are using the existing package Driscoll et al. (2014)[3]. Setting the number of support points comparable to our BP (so that they have similar degrees of freedom), we first run the AAA algorithm for all of our training sets (number of points ranging from 1 to 50) and get the training AAA parameters. Then we used a neural network to model the relationship between the measured magnitude to the AAA parameters for the training data. The network structure we used was 64 neurons with 2 hidden layers with Relu activation. After the network is

---

[3]We used this public Python version: https://github.com/c-f-h/baryrat

properly trained (reach convergence for 300 epochs), we input the testing spectra magnitude and get the inferred AAA parameters for the test spectra. Then we simply reconstructed the barycentric representation and compared the reconstructed phase function with the ground truth to get the MSE measure.

As graphs in the main text suggested, all of the AAA networks performed poorly compared to BP and to naive NN results. This is expected due to following reasons:

1. The functional form of barycentric rational functions lacks actual physics meaning: Encompassing all (m, m) rational functions, they are too large and not restrictive enough for the phase function. The lack of actual physical meaning in the rational approximation unlike our Blashke product (which takes into account the unit circle property of phase functions) is the most important reason we suspect this not working

2. The two-stage process error: There are two stages in our AAA-network baseline, a AAA parameter fitting stage and a network mapping learning stage. The error of each stage would compound and damage the overall performance

3. One-to-many issue: Unlike the Blashke product where the parameters form an ordered set, the parameters of the barycentric representation of AAA form an unordered set. The order is important for neural networks to learn a meaningful mapping as it would not have one-to-many problems, which damages the training performance

### A.6    BASELINE MODEL 3: MUSIC ALGORITHM DETAILS

Multiple Signal Classification (MUSIC) algorithm Schmidt (1986b) is a signal processing algorithm popular in frequency finding and direction finding where the signal is linear combination of multiple signal source of different frequencies and/or directions. MUSIC assumes that the signal vector is consist of p complex exponentials whose frequencies w are unknown.

Since MUSIC is not originally applied to phase retrieval problems and there is no widely single way to do phase retrieval, we would describe fully the process of our MUSIC phase retrieval algorithm here. First, we would assume that our signal $x$ is consist of the sum of p complex exponentials with unknown frequency w and noise term n:

$$x = As + n$$

$$A = \begin{pmatrix} 1 & 1 & ... & 1 \\ e^{jw_1} & e^{jw_2} & ... & e^{jw_p} \\ e^{j2w_1} & e^{j2w_2} & ... & e^{j2w_p} \\ ... & ... & ... & ... \\ e^{j(M-1)w_1} & e^{j(M-1)w_2} & ... & e^{j(M-1)w_p} \end{pmatrix}, s = [s_1, ..., s_p]^T$$

where A is the Vandermonde matrix of steering vectors and s is the amplitude vector. In the assumption the actual number of sources is much lower than the observation, therefore p ¡ M. With this assumption, we would make another assumption that the signal comes from the same distribution, namely their autocorrelation matrix (M by M) should be a constant. Therefore, we would estimate this autocorrelation using the average of the sampled correlation matrix:

$$\hat{R}_x = \frac{1}{N} X X^H$$

Now with the estimate of autocorrelation, we estimate the frequency of this matrix using the eigenspace method, where the eigenvectors associated with a small eigenvalue would be treated as the noise subspace and would be used to recover the signal frequency. Specifically, we decomposite the $\hat{R}_x$ into eigenvalues and eigenvectors, took the smallest (M-p) eigenvalues and their corresponding eigenvectors to form noise subspace $U_N$. Since the eigenvectors from a Hermitian matrix should be orthogonal to each other, meaning any pure signal from signal space would be orthogonal to all the vectors that span the noise subspace. Using a squared norm distance, for a signal $e$, the distance:

$$d^2 = ||U_N^H e||^2 = e^H U_N U_N^H e = \sum_{i=p+1}^{M} |e^H v_i|^2$$

should be zero for all pure signals. Or, the inverse of $d^2$ should be the peaks in the $w$ space when we sweep through a range of possible $w$ values. Now, we sweep the $w$ space with a fine grid and the peaks in the $\dfrac{1}{d^2}$ would be our $w$ values. Therefore, we can construct the A matrix by the top-p peaks in the curve.

The above is a standard MUSIC algorithm to find the signal frequency (assuming the signal is a sum of exponential), now with the frequency of the signal found, finding the actual signal x would be equivalent to finding the corresponding amplitude vector s. Therefore we constructed our phase retrieval problem as an optimization problem:

$$x^* = A \arg\min_s \left( ||\hat{x}| - |x_{gt}||^2 \right) = A \arg\min_s \left( ||As| - |x_{gt}||^2 2 \right)$$

Where the $|x_{gt}|$ is the ground truth magnitude or experimental measurements. For each test phase retrieval problem (each test magnitude spectra), we would need to run the optimization again using the estimated A matrix. We tried the Newton raphson method for this operation but it did not converge even if we set a larger tolerance. Therefore we used the PyTorch framework (without any neural network) and utilized the parallel, GPU-based gradient optimization ability. We constructed the computational graph using the above equation and randomly initialized 1024 initial guesses and used Adam optimizer for 300 epoch till the loss converges. Afterward we would take the best performing single fit among the 1024 candidates as our final answer for this single magnitude. Due to time constraints (300 epoch with 1024 initialization for each of our test cases), we did 500 test cases instead of the full dataset like other methods.

## A.7 DETAILS ON METAMATERIALS DATASET

**Metamaterial absorber:** The input parameters include the four geometric dimensions of the top metal layer $a, w_1, w_2, g$, material properties of a top dielectric layer $\epsilon, \tan\delta$, the conductivity of the metal $\sigma_{metal}$, and thicknesses for all layers $d_{cap}, d_{sub}, d_{m1}, d_{m2}$. The output is the complex reflection coefficient $r(\omega)$, with 1000 frequency points for each of the real and imaginary part of $r(\omega)$ evaluated from $0 - 200$ GHz.

**Membrane Metasurface:** The metasurface supercell geometry has a periodicity $p$, including four elliptical SiC resonators with consistent height $h$. Each elliptical resonator has a major axis radius $r_{ma}$, a minor axis radius $r_{mi}$, and a rotational angle $\theta$. The fourteen geometry parameters define inputs of computational simulations and yield complex reflection and transmission responses. This ADM dataset is particularly interested in the frequency-dependent reflection and transmission range from 100 - 500 THz.

**Synthetic II: Lorentzian Response Function** The second synthetic dataset is generated by considering a causal model of a non-magnetic material, with a frequency-dependent complex dielectric permittivity defined as a sum of Lorentzian oscillators:

$$\epsilon_r(\omega) = \epsilon_\infty + \sum_i^N \frac{\omega_{\epsilon,p,i}^2}{\omega_{\epsilon,o,i}^2 - \omega^2 - j * \omega_{\epsilon,s,i}\omega}.$$

A simple Lorentzian model of the dielectric function is applicable to a wide range of resonant material systems with response functions that must obey Kramers-Kronig relations. Using the transfer-matrix method, Markoš & Soukoulis (2008), the complex transmission coefficient $t(\omega)$ can be computed directly from the dielectric function via the closed form equation:

$$t(\omega) = \left[ \cos\left(\frac{n\omega l}{c}\right) - \frac{i}{2}\left(z + \frac{1}{z}\right) \sin\left(\frac{n\omega l}{c}\right) \right]^{-1},$$

where $n = \sqrt{\epsilon_r}$, $z = \sqrt{1/\epsilon_r}$, and $c, l$ are pre-defined physical constants. We generate a dataset of complex transmission coefficients $t$ by uniformly sampling Lorentzian parameters for $\epsilon_r$ optimized for the frequency range 100-500 THz. We consider a fixed set of four Lorentzian oscillators, and generate 50 sample transmission spectra for training, and 2000 spectra for the test dataset.

