# OpenReview forum: "Blaschke Product Neural Networks (BPNN): A Physics-Infused Neural Network for Phase Retrieval of Meromorphic Functions"
_ICLR.cc/2022/Conference — ICLR 2022 Poster_

### Official Review · Reviewer_64Gx · 2021-10-30

**Correctness:** 3
**Technical Novelty And Significance:** 4
**Empirical Novelty And Significance:** 3
**Recommendation:** 6
**Confidence:** 3

**Main Review:**

I am not aware of a publication that considers computing Blaschke products. Therefore, if this is a good idea compared to previous approaches, then the manuscript is valuable and novel. However, while the paper compares against different neural network structures and setups, it does not compare to standard approaches for rational approximation or phase retrieval that do not involve neural networks.  In particular, I find that there is a significant lack of motivation of why one would use a neural network to compute a Blaschke product, as opposed to using existing data-driven methods for approximating functions by rational functions or exponential sums. For example, for data-driven methods for approximating functions by rationals we have the AAA method (“The AAA algorithm for rational approximation” by Nakatsukasa, Sete, & Trefethen) as well as data-driven methods for approximating by exponential sums, see Prony’s method and MUSIC.

What is the main motivation for using piecewise BPNN? Is that so the Blaschke products can be kept low-degree?  A potential weakness is that the Blaschke products have a fixed degree as the structure of the neural network is fixed.  In data-driven methods for approximating with rational functions, it is found to be essential to adaptively select the degree of the rational function; otherwise, there are numerical stability issues and spurious poles. Can the authors comment on this?

I am also confused by the numerical results. What is the y-axis in the MSE plots?  Is it the logarithm of the MSE in base 10 or the natural log?

**Summary Of The Paper:**

In complex analysis, the Blaschke product is an important family of bounded analytic functions in the open unit disk that is constructed to have roots at prescribed complex numbers. Finite Blaschke products are classically used for constructing rational functions with prescribed poles or zeros, which are then used in the approximation theory, differential equations, dynamical systems, and harmonic analysis.  Here, the authors use finite Blaschke products for phase retrieval of holomorphic and meromorphic functions. By looking at the argument of the approximant, the authors are able to estimate the phase at a set of frequencies. The authors use an L2 loss function to learn the parameters in the Blaschke product.

**Summary Of The Review:**

I believe this manuscript has a promising idea of using neural networks to learn the parameters inside of a Blaschke product. Since this is fundamentally a technique for approximating functions of one complex dimension, the application to phase retrieval seems very natural. I think the authors have a half-baked idea, though. In my opinion, they fail to fully compare or motivate their choices. In particular, I wonder how the author's approach compares to other rational function approximation schemes that do not use neural networks such as the AAA algorithm.

---

> ### Author Response · Authors · 2021-11-18
> **Including AAA method and MUSIC, and addressing concerns about BPNN**
>
> Thank you for your thoughtful review.  Please find our responses to your feedback enumerated below.
>
> ### Feedback 1: This work should compare the BPNN with the AAA method.
> The AAA method is a fundamental work and works well for rational approximation. **However, it requires samples on the target function**. In our phase retrieval problem, when **the only given information is magnitude(without phase)** at different frequencies, we don’t have the samples required for AAA estimation, despite the phase information we have for phase function of other meta-materials. But we appreciate that the reviewer has pointed us to a different rational approximation method to Blaschke Product and **we have tried to infuse the AAA method into our framework to make a comparison with Blaschke Product Neural Network**. Specifically, we first use the AAA methods to find the parameters for the phase functions in the training dataset, then we train a neural network for the mapping from magnitude to the AAA parameters. For the test phase functions, we use the trained neural network to output the predicted AAA parameters and reconstruct the predicted phase function. Essentially we have replaced Blaschke Product with the AAA method for estimation and representation of the rational function. The results are included in section 4. We can see from the comparison that the **AAA method has performed poorly compared to the Blaschke Product**. And we have some conjectural explanations: (1) While the AAA method can represent a wide range of rational functions, the Blaschke Product focuses on meromorphic  functions of unit norm, and thus the **more compact function space is more suitable for representing phase functions** (as |exp(i \theta)| = 1 for any phase \theta). (2)  The AAA method is based on estimating poles explicitly which **may produce sensitivity in phase retrieval**. (3) the AAA method cannot be combined with a function approximator (e.g. neural network) to have an end-to-end model due to its iterative nature. Splitting the model can cause extra error (fitting error of the AAA method and the mapping error from magnitude to the AAA parameters) and lacks the guarantee that the AAA parameters are a smooth function of magnitude (small changes of magnitude should lead to small changes of the parameters). On the other hand, BPNN can overcome these problems as an end-to-end model (Please see appendix for more details and discussion about the AAA method).
>
> ---
>
> ### Feedback 2: This work should compare the BPNN with Prony’s method and MUSIC.
> We appreciate the reviewer’s helpful advice about comparison to data-driven approaches that use sum of exponentials, and we have included the performance comparison of MUSIC and PBNN in the updated manuscript. **Prony’s method (MUSIC) relies on the pivotal assumption that the function is sparse in the number of basis functions**. Please note that Meromorphic functions are not necessarily sparse so the Prony’s method and other  sparse phase retrieval methods may not directly apply. However the MUSIC is more advanced than Prony’s method and we agree that it can be an appropriate baseline.
>
> ---
>
> ### Feedback 3: What is the main motivation for using piecewise BPNN? A potential weakness is that the Blaschke products have a fixed degree as the structure of the neural network is fixed. In data-driven methods for approximating with rational functions, it is found to be essential to adaptively select the degree of the rational function; otherwise, there are numerical stability issues and spurious poles. Can the authors comment on this?
> Yes. High degrees in polynomials can cause **numerical inaccuracies**(on a digital computer) for extremely high frequencies (as is the case for meta-materials). Also piecewise BPNN can also enable finer controls over the complexity of estimation. Some frequency ranges have less complexity than the others and we can **enforce this prior knowledge by choosing the right split and number of roots for piece BPNN**. While we agree that adaptively choosing degrees is important, we believe that the training phase functions and the test phase functions should have **homogeneity of complexity**. Hence the configuration of BPNN that works well on the training dataset should also generalize to the test dataset. Another important point is that, based on empirical results,**BPNN learns to zero out extra unnecessary degrees of complexity on training datasets and generalize well on test datasets** (Please see Appendix for our study of effects of hyper-parameters).
>
> ---
>
> ### Feedback 4: What is the y-axis in the MSE plots? Is it the logarithm of the MSE in base 10 or the natural log?
> We apologize for the confusion caused by the labels of y-axis. They are given in dB (decibels) with **logarithm in base 10**. We have updated the descriptions in the paper.

---

> ### Comment · Reviewer_64Gx · 2021-11-27
> **Response to the authors**
>
> I wish to thank the authors for their response to my review. I welcome the comparisons to the other methods that I mentioned, including MUSIC and AAA. I am happy to update my score.

---

### Official Review · Reviewer_9QvZ · 2021-11-02

**Correctness:** 4
**Technical Novelty And Significance:** 3
**Empirical Novelty And Significance:** 3
**Recommendation:** 8
**Confidence:** 3

**Main Review:**

I would suggest to conduct a few additional experiments to better analyze the model behavior. First, a comparison against other baselines than the neural network model should be performed if feasible, e.g. compressed sensing based methods or even the Kramers and Kronig formula. While the latter might not be a good choice due to the problems with the Riemann sum approximation for few datapoints, it could e.g. still provide a more complete picture.  It would further be interesting to see, how the methods behave for larger training/test set splits (e.g 50:50). Similarly, the influence of different possible configurations of BPNNs is underexplored. How do they behave with respect to the number of roots used? What is the influence of different window choices on the piece-wise BPNN implementation? Regarding the results section in general, the paragraph on the MMA experiments (page 8) is hard to follow and should be reworked. In general, including only the learning curves of a few representative NNs in the figures should be sufficient, the rest could e.g. be moved to an appendix.

The section regarding model architecture would profit from additional information as well. It is, for example, unclear what activation functions were used in the BPNNs and fully connected NNs or what kind of optimizer was used for training the models.

Finally, there is a slight confusion regarding the number of frequencies used in the metamaterial absorber dataset. Table 1 reports 256, while the text states that 1000 frequency points are used.

**Summary Of The Paper:**

The authors introduce a neural network architecture based on Blaschke products for phase retrieval applications.

**Summary Of The Review:**

In general, the idea to use neural networks to parametrize Blaschke polynomials for predicting phase functions is well motivated and shows promise based on the experimental results. The architecture and theory are presented in a concise fashion. However, the evaluation of the model is quite minimal and could profit from a more in-depth analysis in the form of additional experiments. Nevertheless, I tend towards accepting the work.

---

> ### Author Response · Authors · 2021-11-18
> **Kramers and Kronig added, ablation study added and clarity enhanced**
>
> We are grateful for your comments and your helpful advice that improved our paper. We have included two extra baselines, **Kramers-Kronig Method** and **MUSIC algorithm**(an algorithm suggested by another reviewer that estimates the function with sum of exponentials on the premise that the function is sparse) for a more complete picture. **Our experimental results suggest that their performance is substantially worse than BPNN’s**. In Section 4.3 we postulate explanations for the observed performance differences among the models.
>
> We also agree that some insights for hyper-parameters of BPNN are useful for users of BPNN, and **we have studied the effects of number of roots and number of windows (which changes size of windows) on performance of BPNN and included it in our appendix**.
>
> We apologize for the confusion caused by the conflicting information and MMA result paragraph. We have fixed the conflicting frequency number, re-written the MMA results paragraph and included more details about BPNN (Specifically, we used **ReLU activation function** and **Adam** for training of all neural networks and BPNNs. We appreciate your recognition that this important information is missing).

---

> > ### Comment · Reviewer_9QvZ · 2021-11-19
> > **Updated recommendation**
> >
> > I believe that the changes to the paper, especially the comparisons with other methods, have greatly improved its quality. I have revised my score accordingly.

---

### Official Review · Reviewer_kQWk · 2021-11-02

**Correctness:** 3
**Technical Novelty And Significance:** 3
**Empirical Novelty And Significance:** 2
**Recommendation:** 5
**Confidence:** 3

**Main Review:**

**Strong Points**

- Imposing priors on neural networks based on the physics of the problem is very important. Using Blaschke products to do this is an interesting idea.


**Weak Points**

- No comparison to existing methods in phase retrieval.
- No ablation study.


**Questions**

- How does the size of the Blaschke product in Theorem 1 depend on epsilon? How do you choose the degree of the rational function?
- Why is a neural network used to predict the coefficients for the rational function? How does this compare to simply fitting the ration function directly?
- In Section 4.1.1, p(z) is “randomly selected.” What does this mean? What is the distribution over p(z)?


**Additional Feedback**

- Section 4.2: “Each NNs are trained with 6000…” This is still no guarantee of convergence. Other metrics, like gradient norm should be monitored for this.
- There are many small grammatical errors.


**Summary Of The Paper:**

**Summary**

The paper proposes a new method, BPNN, for phase retrieval, where the phase of some holomorphic or meromorphic function is predicted from measurements of its magnitude. The authors show that the unknown function can be expressed in terms of Blaschke products. Given some tolerance epsilon to the unknown function, the Blaschke product amounts to a rational function. A neural network is then used to predict the coefficients of this rational function and is trained using L2 loss. Finally, BPNN is compared against using a neural network to directly approximate the unknown function.

**Summary Of The Review:**

**Recommendation**

My sense is that in its current form the paper should be rejected. Especially given the lack of comparison to existing methods. I am not an expert on phase retrieval, so I am happy for the other reviewers and rebuttal phase to change my mind.

---

> ### Author Response · Authors · 2021-11-18
> **Addition of four extra baselines**
>
> Thank you for your thoughtful review. Below please find our responses to your feedback.
>
> ---
>
> ### Feedback 1: No comparison to existing methods in phase retrieval.  No ablation study.
> We agree that a comparison with additional existing methods, and associated ablation studies, would strengthen our work. Therefore, based upon the reviewer’s feedback **we have included more comparisons as well as ablation studies**. For three meta-material datasets, we have included comparison to
> 1. **Kramer-Kronig Method**: the conventional method for phase retrieval with only magnitude methods;
> 2. **MUSIC algorithm**: phase retrieval method that uses sum of exponentials as an approximation for phase functions;
> 3. **Linear+BP**: replacing the neural network in BPNN with a linear regressor
> 4. **AAA method**: a different rational approximation algorithm to Blaschke Product(see more about the AAA method in the appendix).
>
> We believe that the presented methods provide a comprehensive set of baseline comparisons and also our ablation studies are representative. **These additional results can be found on Figs. 4, 5 and 6 of the revised manuscript.**
>
> We briefly summarize the new results here, although a more detailed discussion can be found in Section 4.3 of the revised manuscript.  As it can be seen in our updated plots and appendix, (1) The KK method and MUSIC perform poorly because  they may make important assumptions about the target functions that for the missing phases may not necessarily hold. Also the KK method requires available data in a wide range as it requires one to evaluate an integral over the whole frequency axis. (2) While the AAA method can represent a wide range of rational functions, the Blaschke Product focuses on meromorphic functions of unit norm, and thus the more compact function space is more suitable for representing phase functions (as |exp(i \theta)| = 1 for any phase \theta). (3) The AAA method is based on estimating poles explicitly which may produce sensitivity in phase retrieval. (4) the AAA method cannot be combined with a function approximator (e.g. neural network) to have an end-to-end model due to its iterative nature. Splitting the model can cause extra error (fitting error of the AAA method and the mapping error from magnitude to the AAA parameters) and lacks the guarantee that the AAA parameters are a smooth function of magnitude (small changes of magnitude should lead to small changes of the parameters). On the other hand, BPNN can overcome these problems as an end-to-end model.
>
> ---
>
> ### Feedback 2: How does the size of the Blaschke product in Theorem 1 depend on epsilon? How do you choose the degree of the rational function?
> This is an important and intriguing question about the relationship between the precision and the required number of degrees in the Helson-Sarason Theorem. Although Dirichlet proved that the accuracy of estimation a of a given irrational number with a fraction of the form p/q (p and q are integers) grows quadratically with $q^2$,  we are not aware of such a similar result for the Helson-Sarason Approximation Theorem discussed in the paper. **In fact, the required degree for  Blaschke Product to reach a specific $\epsilon$> 0 accuracy is extremely difficult to quantify. However, from our empirical study with a comprehensive list of data sets, Blaschke Product can achieve high precision with a decent number of degrees,** and piecewise BPNN can further parallelize the estimation and improve numerical stability by decreasing the highest degree number in each frequency sequence.
>
> ---
>
> ### Feedback 3: Why is a neural network used to predict the coefficients for the rational function? How does this compare to simply fitting the ration function directly?
> Unfortunately we cannot use Blaschke Product to fit an individual observed function. **Because the only information we have in the phase retrieval problem is magnitude and we need truth phase information to do fitting.** Although representing the phase function with a Blaschke Product has already decreased the complexity of the prediction problem, **the problem of mapping the magnitude to Blaschke coefficients is non-linear**. To demonstrate this, we have included in our appendix the performance of using linear regression instead of neural network to predict Blaschke coefficients for phase functions.
>
> ---
>
> ### Feedback4: In Section 4.1.1, p(z) is “randomly selected.” What is the distribution over p(z)?
> The roots of the polynomials are sampled **uniformly** on the unit circle. We have included this detail in our updated manuscript(section 4.1.1).
>
> ---
>
> ### Feedback 5:  Section 4.2: “Each NNs are trained with 6000…” This is still no guarantee of convergence. Other metrics, like gradient norm should be monitored for this.
> All the neural networks in the experiments have converged after 6000 epochs and we have updated our paper to include this important fact in section 4.2.

---

> > ### Comment · Reviewer_kQWk · 2021-11-26
> > **Updated score**
> >
> > I am grateful to the authors for their additional work. The comparison to existing baselines, including linear regression on the coefficients, has certainly improved the paper, so I will increase my score.

---

### Official Review · Reviewer_cH7K · 2021-11-05

**Correctness:** 3
**Technical Novelty And Significance:** 3
**Empirical Novelty And Significance:** 3
**Recommendation:** 6
**Confidence:** 4

**Main Review:**

I liked a number of aspects of this paper. I thought the problem was well-motivated and the authors did a good job of explaining the underlying theory. Although I am not an expert on phase retrieval, the proposed method seems pretty elegant and clever. The experiments also seemed relatively thorough and the authors definitely tried their method on a number of different problems.

Having said this, there are a number of ways in which I think the exposition could be improved (it is also possible that I am missing some details).
1. Reading the paper, my main question was what the fully-connected neural network brings to the table? If I understand correctly, it seems like the authors could have just regressed the coefficients of the rational approximation directly. I’d appreciate some clarity on why this couldn’t be done or why it wasn’t included as a baseline.
2. I think I’m missing something about the training procedure. In particular, the equation for the backward pass features f(j\omega_i), which presumably includes phase information. However, I thought the premise was that only |f(j\omega_i)| was available. Can the authors clarify that this equation is really correct as written?
3. I think the section on the piecewise BPNN could use some elaboration since it wasn’t totally clear what was happening here.
4. I, personally, could have used a refresher about why the function values are only available on the imaginary line.
5. While it is nice that the authors included code (which did help me understand exactly what was happening) I think the authors should clean it up a bit and add some comments before the conference.
6. There were a number of places in the paper where the authors used terminology such as “it is easy to see that…” Typically I don’t find this style of exposition especially helpful. In particular, I think a little bit more detail on the rational approximation to B_k(j\omega) could have been nice.

Despite these issues, I do think this paper seems interesting and potentially useful. I also think the trick of using Blaschke products is something I hadn’t seen before and other researchers might find it interesting. So I do support publication.

**Summary Of The Paper:**

This paper introduces a new neural network architecture, called the BPNN, to recover phase information in physical problems where one only has access to the magnitude of a function on the complex line. This is a common problem in experimental physics and scattering where phase information can be lost, but is nonetheless important. To recover the phase in this setting the authors propose approximating the phase using Blaschke Products, which can approximate any continuous function on the imaginary line whose magnitude is one. Specifically, the authors train a fully-connected neural network to compute the coefficients of a rational approximation to the Blaschke product. They demonstrate the success of their method on a number of example problems and include code to reproduce their results.


**Summary Of The Review:**

A clever paper using a relatively uncommon mathematical trick to make progress on a niche, but important problem. While the paper could be improved, I do think the ideas here could be interesting and useful to the community. The experiments seem relatively compelling and code is included with the submission.

---

> ### Author Response · Authors · 2021-11-18
> **Clarity greatly enhanced**
>
> Thank you for your thoughtful review.  Please find our responses to your feedback enumerated below.
>
> ---
>
> ### Feedback 1: Reading the paper, my main question was what the fully-connected neural network brings to the table? If I understand correctly, it seems like the authors could have just regressed the coefficients of the rational approximation directly.
>
> Unfortunately, we think that Blaschke Products cannot be used for producing a direct fit to test data in an obvious manner because **Blaschke Products estimates the phase function while for the test data in the phase retrieval problem only the magnitude information is given.**  In order to perform a direct fit, we need at least some measurements of true phase information at some frequency points (but this may be very difficult to obtain). Hence, we have taken a data-driven approach. By representing the complete phase function as Blaschke product instead of making predictions of phases at thousands of frequency points, the complexity of the problem is reduced. Please kindly note that despite this reduction of complexity, **the mapping from magnitudes to Blaschke Product is still non-linear**. To demonstrate this, **we have added into the paper the performance of using a linear regressor, instead of a neural network, for prediction of Blaschke coefficients.** As it can be seen in the updated manuscript, the result is not very good, but it is still better than only using classical neural networks in some datasets.
>
> ---
>
> ### Feedback 2: I think I’m missing something about the training procedure. In particular, the equation for the backward pass features f(j\omega_i), which presumably includes phase information. However, I thought the premise was that only |f(j\omega_i)| was available.
>
> It is totally correct that only magnitude information |f(j\omega_i)| is observed for phase retrieval problems. But we assume we will have the true magnitude + phase information  f(j\omega_i) for our training data.
>
> ---
>
> ### Feedback 3: Refresher about why the function values are only available on the imaginary line.
>
> Thank you for your question about why the function values are only available on the imaginary line. **We believe this is a valuable question for readers unfamiliar with the phase retrieval problems and have included the following explanation in the appendix for readers who also have this question.** In physics, electrical engineering and other fields, solutions to many linear PDEs with constant coefficients, and also many other measurements are done in the frequency domain.  This in principle is achieved by multiplying a real signal x(t)  by sinusoidals $cos(\omega t)$ and $\sin(\omega t)$, and integrating over time (Calculating Cosine and Sine Transforms) for various values of $\omega$.  If x(t) is a real physical signal, these cosine and sine transforms are real and imaginary parts of $X(j\omega)$, the complex Fourier transform of x(t). Thus $X(j\omega)$ gives the value of $x(\cdot)$ over the imaginary axis.
>
> ---
>
> ### Feedback 4: More explanation about piecewise BPNN and cleaning codes
>
> Thank you for your suggestion for revising the exposition of the paper. **We have added more explanations about our theory and more details about piecewise BPNN**. As you suggested, we will add more comments and package our BPNN as API for easy usage. In fact, we will **include codes for two other phase retrieval methods**- KK methods and MUSIC that we have just included in the paper as two extra baselines.

---

### Author Response · Authors · 2021-11-18
**Four extra baseline methods and addition of details about BPNN as well as the theories**

Thank you for taking the time to review our manuscript and provide feedback - we believe that your feedback has greatly improved our work!  Based upon the reviewers’ feedback, we have made the following major changes to our papers, and we have uploaded a revised manuscript for your consideration:

1.**Including four extra baseline methods** and they are summarized in the following table:

| Baseline | Data-Driven | Function Approximation | Description |
| --- | --- | --- | --- |
| Kramer-Kronig Method | No | *NA* | Conventional phase retrieval method that uses causality |
| MUSIC algorithm | Yes | Sum of exponentials | Data-driven function approximation method that estimates functions as sum of exponentials |
| AAA-network algorithm | Yes | Barycentric rational functions | AAA[1] is an established rational approximation technique. Similar to BPNN we use its functional form to approximate the phase function |
| Linear + Blaschke Product | Yes | Blaschke Product rational functions | Using linear regression instead of neural networks in BPNN |

We believe the above four extra baseline methods on our inclusive set of datasets make a comprehensive set of baselines and thus a thorough ablation study. In summary, the additional experiments reveal that **the BPNN achieves superior accuracy compared to all existing approaches now considered. We believe these additional results strengthen our conclusions, and added substantial value to the work**. We also include in our paper: **details about the baseline methods**(in appendix) as well as **our postulations about reasons of unsatisfactory performance of the baseline methods**(short version in paper and detailed version in appendix)

2.**Remake all plots** as well as their captions for clarity

3.**Addition and clarification of details** about BPNN structure, NNs, training methods, y-axis values as well as more details in our theories and properties of phase functions.

4.**Study about hyper-parameters** of BPNN (in appendix). In particular, we find that BPNNs are robust to overshooting of degrees.

Answers to each reviewer’s specific questions have been posted individually. We think the questions and comments are extremely helpful and we welcome any further suggestions.

Thank you again,

Authors.


[1] Nakatsukasa, Yuji et al. “The AAA Algorithm for Rational Approximation.” SIAM J. Sci. Comput. 40 (2018): n. pag.

---

### Decision · Program_Chairs · 2022-01-20

**Decision:**

Accept (Poster)

**Comment:**

The Authors propose a neural-network based approach for the phase retrieval problem. Solving the phase retrieval problem is key for important application areas such as crystallography or radioastronomy.

After adding more baselines and other changes, 3 out of 4 reviewers recommended acceptance. Reviewer kQWk recommended rejection mostly based on the fact that the paper is quite narrow in scope.

Reviewer kQWk is right that the topic might not appeal to most of the ICLR community. It is worth noting that the main contribution of the paper is not about neural networks but rather about connecting phase retrieval with Blaschke products. As it stands, it seems that after making this connection, any non-linear approximator could do well.

Having said that, this is an important application area and the progress is welcomed. Hopefully, it will draw inspire more research in this area.

Currently, the key issue of the paper is that it is very challenging to understand for people without background in the phase retrieval area or complex analysis. To make this paper more valuable for the ICLR community, I would strongly encourage the Authors to devote at least a page to explanation of what is the phase retrieval problem, and the intuition behind the solution. Perhaps [1] could serve as an inspiration.

[1] Phase retrieval in crystallography and optics, R. P. Millane